# Improvement of violence management among nurses in Iran: The best practice implementation project in a health promoting hospital

Ghasem Abedi[1], Elaheh Haghgoshayie[2]*, Edris Hasanpoor[2], Jalil Etemadi[3], Morteza Nazari[4], Razieh Vejdani[5]

1 Health Sciences Research Center, Mazandaran University of Medical Sciences, Sari, Iran, 2 Research Center for Evidence-Based Health Management, Maragheh University of Medical Sciences, Maragheh, Iran, 3 Department of Clinical Sciences, Maragheh University of Medical Sciences, Maragheh, Iran, 4 Department of Healthcare Services Management, School of Health, Alborz University of Medical Sciences, Karaj, Iran, 5 Assistant Professor of Obstetrics and Gynecology, Shahid Beheshti Hospital, Maragheh University of Medical Sciences, Maragheh, Iran

* ezd_ehm2010@yahoo.com

**Data Availability Statement:** Data cannot be shared publicly as the informed consent we obtained from our participants limits our ability to

## Abstract

### Background

This project aimed to assess compliance with evidence-based criteria' for the prevention and management of workplace violence against nursing staff in Shahid-Beheshti hospital, Maragheh, Iran. Workplace violence is a managerial and workplace occupational health and safety issue that can affect the performance of an institution. Further, it might turn the work environment into an insecure and hostile one which can influence the performance of employees and their professional relationships negatively. Nevertheless, staff have their own legal rights, and their organizations are legally and ethically in charge of providing them with safe work environments.

### Methods

Following the JBI Practical Application of Clinical Evidence System and Getting Research into Practice audit and feedback tool with three phases of activities, this project utilized an implementation framework incorporating quality improvement. Furthermore, the audit tool was used to establish the project and set up the measurement and evaluation of three evidence-based criteria.

### Results

The post-implementation audit results indicated a significant improvement in violence management and prevention. The compliance rate on the first criterion, i.e. aggression management training, increased from 49% at baseline to 81% at the end. The second criterion, i.e. timely support and assistance following an incident, exhibited greater increase from eight to

share the raw data collected. There is some sensitive information contained within some of the transcripts which potentially can be identified. The Research Ethics Committee of Maragheh University of Medical Sciences, Maragheh, Iran imposes ethical restrictions in sharing data. However, data request may be sent through the Research Ethics Committee of Maragheh University of Medical Sciences by maragheh.research@yahoo.com for researchers who meet the criteria for access to confidential data.

**Funding:** The author(s) received no specific funding for this work.

**Competing interests:** The authors have declared that no competing interests exist.

73%. Finally, an increased compliance was noted on the third criterion, i.e. policy for risk management and safe environment, from 22 to 77%.

## Conclusions

The current project successfully implements evidence-based violence management in Sha-hid-Beheshti hospital. It reveals significant results on compliance and the increasing knowl-edge of nurses on evidence-based stress management, communication skills and self-companion.

## Introduction

Violence, as one of the most important issues in public health, is experienced by most people [1, 2]. Violence occurring in workplace is called workplace violence (WPV). It is defined as any act in which a person is abused, threatened, intimidated, or assaulted during his/her employment [3, 4].

Similarly, in healthcare, WPV against and among healthcare workers consists of abuses, blame, intimidations, and circumstances related to work. It is undeniable that healthcare pro-fession is potentially hazardous; it can be due to the proximity to disease and pathogens, han-dling chemicals or sharp tools, or even slips and falls [5]. However, WPV has overstepped all the other risks and is a significant occupational hazard for healthcare workers [6]. Indeed, healthcare workers are the most vulnerable professionals to WPV [7].

More than 50% of healthcare workers are exposed to violence, which makes it a serious problem in healthcare systems [8, 9]. Recent studies show that approximately 20% of violent incidents occur in healthcare settings and healthcare staff experience violence 16 times more often than others do [10, 11]. WPV has numerous negative physical and psychological conse-quences for healthcare workers, e.g. leading to decreased job motivation, burnout, depression, and a tendency to quit the job. These outcomes in turn affect the quality of care and hence put health-care provision at danger, not to mention the rise in absenteeism in the workplace or enhancement costs, litigations, and workers' compensations [4, 6, 8–10].

It should be noted that the risk of experiencing workplace violence varies considerably according to occupation. In this regard, nursing holds the first place out of 71 evaluated jobs [11]. A recent systematic review and meta-analysis of studies on WPV against healthcare pro-fessionals found nurses to have the highest prevalence of WPV of any form [12]. Nurses are 7.9% more likely to experience assaults and threats annually [13].

Several lines of evidence have reported that more than half of the nurses in European coun-tries have experienced some type of violence in the workplace. However, the rate of WPV is higher in Asian and Middle Eastern countries, where more than two thirds of nurses were exposed to violence [14–16]. The annual prevalence of workplace violence against nurses was reported 9.5% in the United Kingdom [17], 17% in Canada [18], 36.4% in Japan [19], 44.6% in Hong Kong [20], 91.4% in Jordan [21], 67.4% in Saudi Arabia [22] and 85.2% in Turkey [23]. Also, a 12-month study by Nikathil (2017) revealed that 95% of nurses in Australian hospitals had encountered verbal aggression several times [24]. Chunyan Song (2021) showed that a total of 172 nurses (64.7%) experienced violent incidents during the past year in china [25].

Evidence has also shown that nurses in emergency departments (ED) are most exposed to violence than those working in other departments. It is because compared to other healthcare settings, EDs are high-risk settings for WPV [26]. Nikhil reported that the pooled proportion

of 36 violent episodes per 10000 presentations demonstrated substantial exposure of ED staff to WPV [24].

As mentioned, nurses play a central role within healthcare teams, and frequently work at the front line interacting with all kinds of patients. Recent estimates have reported that 48% of graduating nurses are fearful of workplace violence and more than 60% of all new nursing graduates leave their first job due to negative workplace behavior directed at them [27]. Multi-factorial factors aiding workplace violence include the high-stress environment, care of acutely agitated patients (e.g. intoxicated and psychiatrically unwell patients), longer waiting times, patients with high expectations (unmet expectations), cultural differences, poor funding, high workload, staff shortage, and inadequate security staffing. This, in turn, can generate dissatisfaction, frustration, and a feeling of inequity, all of which lead to an escalation in violence [28, 29]. Also, lack of trained security staff, 24-hour access, very high levels of stress and anxious companions are some of the reasons why this environment is prone to violence [23].

Since the causes of violence are a multifaceted issue, prevention strategies of violence against nurses should be planned and implemented accordingly [30]. Therefore, many countries have taken remedial actions to manage, improve, and reduce WPV [31]. For example, in May 2020, Sudan agreed to establish hospital security teams. India has also approved a law to imprison those who commit violence against medical personnel up to seven years [31]. The studies have also shown that job stress is one of the factors that increases violence in hospitals, the occurrence of which can be reduced by psychological support and stress reduction programs for nurses [32]. Hamedan et al., [33] in Palestine showed that improving the security of emergency departments through increasing security personnel, the presence of alarms and closed-circuit video surveillance can help control violence [33]. Also, Al-shammari et al. found that increasing the number of security staff and improving security systems were helpful for violence control [34]. An Australian study showed that having security guards in the ED, who are a visible attendance and who replied quickly to incidences of verbal abuse and physical assault, helped staff to feel safe [35].

Workplace violence (WPV) against health care workers has been a continuous problem in the worldwide for decades and more studies reported that the incidence of violence against healthcare workers, especially nurses, has increased over the past decade [36–40]. Based on these cases, Khan stated that the violence against nurses is a silent epidemic that has serious consequences [41]. This silent epidemic becomes even more widespread during pandemics and crises. Nurses are often the front-line aid/relief workers in crises. Coronavirus disease (COVID-19) is one of the most recent pandemics in the world. Based on studies, COVID-19, as a pandemic, has severely shocked the health care systems in most countries. The lack of adequate medical and care personnel, particularly nurses, is a major challenge in dealing effectively with this disease, forcing nurses to work longer hours by scarifying many individual plans and leisure activities [42].

Unfortunately, despite great efforts of the clinical team in this crisis, nurses and all the clinical staff are still facing violence in many countries. Rodriguez and colleagues showed that healthcare workers often experience violence. In addition, their results indicate that the rise in COVID-19 cases caused a wave of violence against health workers, who have been wrongly accused of spreading the disease. Their observations have shown that many patients intentionally cough or spit on the clinical staff [43, 44].

The studies conducted in Iran reveal the level of occupational violence against and among nurses between 68.8 to 98.6 percent, notable numbers [11, 45]. Salimi's (2007) study on Violence against nurses in non-psychiatry emergency wards in Iran showed that 97.8% of nurses experienced verbal violence and 39.7% physical violence [46]. Another study in hospitals of East Azerbaijan province reported different types of violence toward nurses in order of

decreasing prevalence as verbal, physical, racial and sexual violence [47]. Therefore, due to the importance of workplace violence and its destructive effects, the project team tried to investigate the causes of violence and provide strategies in Shahid-Beheshti educational hospital [the health promotion hospital]. in Maragheh. The health promotion hospital is a public health facility in the study area providing primary health care services, including vaccinations, for the population in the study area free of charge regardless of their legal migrant status.

## Objective(s)

The aim of this project was to assess current practices, determine and identify strategies to overcome barriers, and implement best practices to improve aggression management among nurses in Shahid-Beheshti hospital. Therefore, the project tried to amend aggression and staff injury by improving staff knowledge. The specific aim of this project was to assess compliance with evidence-based criteria regarding aggression management amongst nurses:

- To identify and engage a multidisciplinary team to determine current compliance with evidence-based criteria regarding workplace violence against nurses.

- To reflect on the results of the baseline audit by following the JBI and GRiP framework, identifying barriers and facilitators of designing and implementing strategies, which address any noncompliance observed at baseline audit.

- To conduct a follow-up audit to assess the results of the interventions implemented, in order to improve violence management against nurses in Shahid-Beheshti hospital.

## Materials and methods

### Design

This project was implemented in three phases over 11 months, from January 2020 to November 2020. The baseline audit was the data collected by the project leader which began on May 2020. The project used the Joanna Briggs Institute Practical Application of Clinical Evidence System (JBI PACES) and Getting Research into Practice (GRiP) audit, and feedback tool for data collection and analysis [48]. The project steps are shown in Fig 1.

### Setting and participants

Since that nurses are frontline activists of the health care system and have the closest contact with patients and their relatives, the project team decided to selected nurses as a project sample. The project was conducted from January 2020 to November 2020 with the audits conducted in Shahid-Beheshti hospital [health promoting hospital; a hospital where the health promotion hospital plan has been implemented.] in Maragheh University of Medical Sciences. The sample size consisted of 86 nurses at pre-audit and 62 nurse at post-audit admitted during the audit period. Sample calculator was used to calculate the sample size of the study.[34] Shahid-Beheshti is a general hospital with approximately 112 beds and 114 nurses. This hospital provides midwifery services, Para-clinic, accident and emergency (A&E).

### Ethical considerations

The study was approved by ethical committee of Maragheh University of Medical Sciences (Ethical code of project: IR.MARAGHEHPHC.REC.1398.053). Also, an approval from the

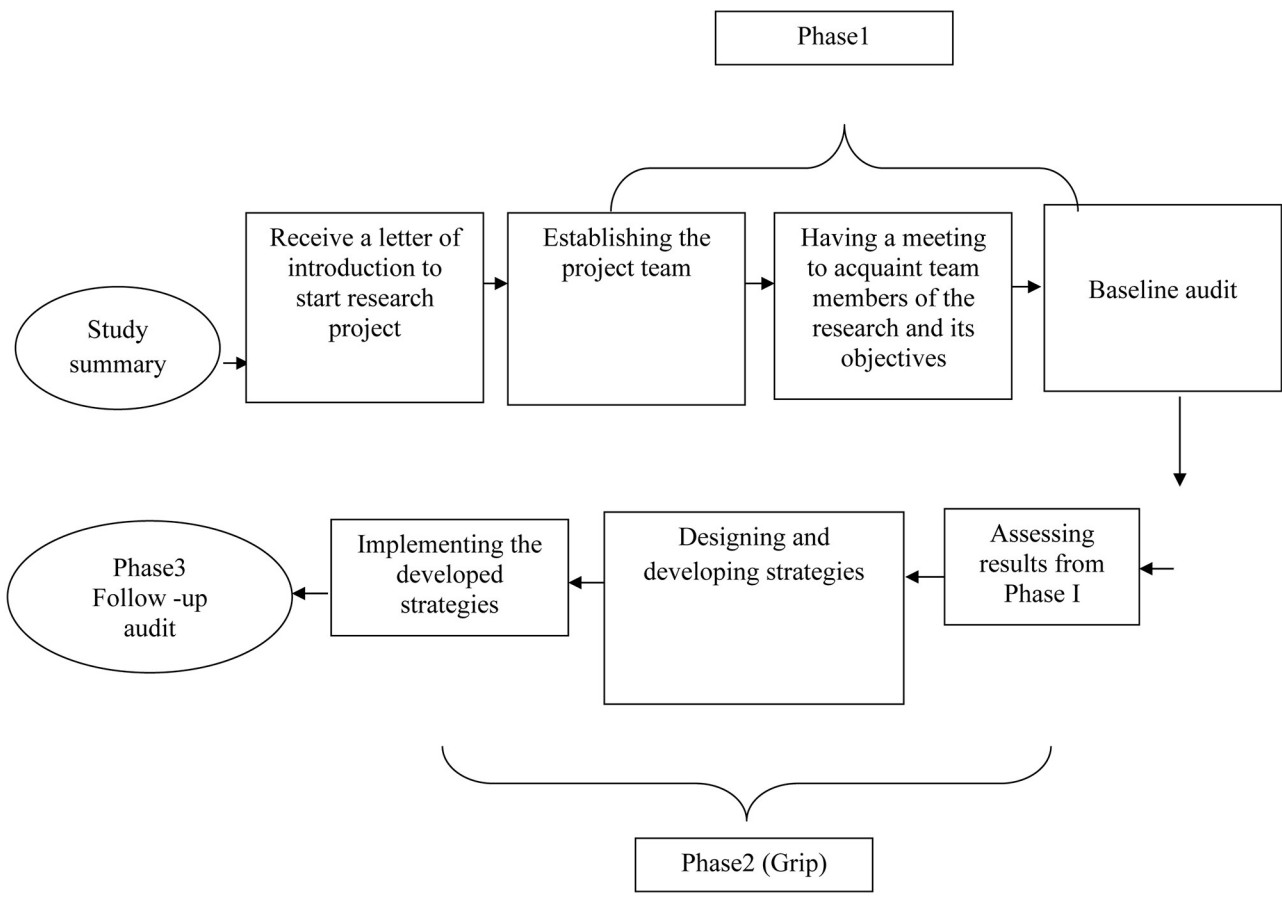

**Fig 1. The project steps.**

hospital ethics committee was acquired. All the participants taking part in this study gave an informed consent and a verbal consent form was completed by all the participants.

## Phase 1: Team establishment and baseline audit

**1–1. Establishing the project team.** The project leader chose the project team based on the experience and position within the facility (Table 1). They were invited to participate in the project based on their positive approach and ability to influence staff and to engage patients. The project team included senior research fellow, chief executive officer (CEO), head nurses, and matron. Involvement of the project team was based on their roles in support, data collection, data entry and participation. The project leader was responsible for managing the team, the project, and the timeline. The role of the team members was as follows:

- Contributor of ideas, facilitator at in-service, auditor and collator of audit criteria

- Implementers of best practice strategies

**1–2. Baseline audit.** Audit criteria were provided by JBI and the criteria were based on the best available evidence presented in existing JBI evidence summaries [49]. Table 2 contains the evidence informed audit criteria used in the project (baseline and follow up audit) together

**Table 1. The members of team.**

| No. | Name | Role in hospital |
|---|---|---|
| 1 | Dr Elaheh Haghgoshayie | Assistant professor in healthcare management, educational deputy of Shahid-Beheshti hospital |
| | | Team lead, gathering results related to meetings, designing and developing strategies, submit report |
| 2 | Dr Ahmad Navinsalari | Hospital manager |
| | | Designing and developing strategies, supporting the implementation of the strategies, the study of legal strategies |
| 3 | Mrs Asadi | Hospital Metron |
| | | Designing and developing strategies, supporting the implementation of the strategies, performance monitoring of strategies |
| 4 | Mrs Gholamhoseinpour | Head nurse |
| | | Designing and developing strategies, identifying effective factors and barriers to the use of evidence, monitoring of implementing strategies, supporting the implementation of the strategies, submit report |
| 5 | Mrs Golanbari | Head nurse |
| | | Designing and developing strategies, identifying effective factors and barriers to the use of evidence, Provide reports on implemented programs, monitoring of implementing strategies in her department, Submit report |
| 6 | Mrs Mohammadi | Supervisor |
| | | Designing and developing strategies, identifying effective factors and barriers to the use of evidence, monitoring of implementing strategies in hospital |
| 7 | Mrs Shekarchizadeh | Head nurse |
| | | Designing and developing strategies, identifying effective factors and barriers to the use of evidence, Provide reports on implemented programs, monitoring of implementing strategies in her department, Submit report |
| 8 | Miss Mehrdad | Head nurse |
| | | Designing and developing strategies, identifying effective factors and barriers to the use of evidence, Provide reports on implemented programs, monitoring of implementing strategies in her department, Submit report |
| 9 | Mrs Bagherpour | Head nurse |
| | | Designing and developing strategies, identifying effective factors and barriers to the use of evidence, Provide reports on implemented programs, monitoring of implementing strategies in her department, Submit report |
| 10 | Mrs Saidisharif | Supervisor |
| | | Designing and developing strategies, identifying effective factors and barriers to the use of evidence, monitoring of implementing strategies in hospital |

with a description of the sample and approach to measuring compliance with the best practice for each audit criterion. The baseline audit started on May 2020 and ended on July 2020. To collect the baseline data, we designed a questionnaire which consisted of demographic characteristics of the respondents (nurses) and audit criteria. The methods used to measure percentage compliance with the best practice included documentation audit and semi-structured interview. The documents included reports of violence recorded in the hospital system. The interviewers included two researchers from the research team. The researchers have previous experience conducting semi-structured interviews of healthcare professionals. The interviews lasted on average for 39 min (range 22–56) for nurses. Additionally, they had been trained to conduct the semi-structured interviews for improving the quality of interviews.

After the project team completed the audit form by reviewing documents and interviews with the participants, the members put the data into JBI PACES and generated results. All the criteria audited were collated by using Yes, No and Not Applicable with the comments section

**Table 2. Audit criteria, sample, and approach to the measurement of compliance with best practice.**

| Audit criterion | Sample | | Method used to measure percentage compliance with best practice |
|---|---|---|---|
| | Baseline Audit | Follow-up audit | |
| Healthcare staff receives aggression management training. | 86 nurses | 62 nurses | By interviewing and reviewing documents related to in-service training programs. |
| | | | Tools: questionnaire, documentation |
| Healthcare staff who experienced violence or aggression receives timely support and assistance following the incident. | 86 nurse | 62 nurses | By interviewing and reviewing documents related to in-service training programs. |
| | | | Tools: questioner, documentation |
| The health organization has a policy for risk management and ensuring a safe environment for all staff and consumers | 86 nurses | 62 nurses | By interviewing and reviewing documents related to in-service training programs. |
| | | | Tools: questioner, documentation |

available to the highlighted areas of concern or in-accuracies of answers [49]. The following three evidence-based criteria were used for the audits:

1-2-1. Healthcare staff receive aggression management training

1-2-2. Healthcare staff who experienced violence or aggression, receives timely support and assistance by following the incident

1-2-3. The health organization has a policy regarding risk management and ensuring a safe environment for all the staff and consumers

## Phase 2: Implementation—Getting Research into Practice (GRIP)

This stage was comprised of three phases: (1) Assessing results from Phase I, (2) Designing and developing strategies, identifying effective factors and barriers to the use of evidence, and implementation of strategies, and (3) Implementing the developed strategies. When assessing the results, findings were provided to all the hospital personnel. In the next stages, group discussion sessions were held to identify the causes of violence, determine strengths and weaknesses, and develop feasible strategies. Given the issue of violence and its effective factors, as well as the cultural barriers and the taboos on talking about violence, the use of both quantitative and qualitative methods could help researchers to acquire rich information about nurses' experiences of violence in their work environment. This method helps to obtain deeper access to the effective factors of violence through interview and meeting sessions and to design and implement more detailed plans and solutions based on the reality of the hospital, as well as cultural and organizational conditions (Phase 2).

Using a mixed-method helps obtain a general overview of violence, its evidence, and the design of evidence-based correctional programs by combining quantitative and qualitative findings. After obtaining baseline audit results in the first phase of this project, focus group discussion sessions were held with the authorities of different units, supervisors, hospital head, matrons, and hospital representatives to develop executive programs and strategies. Focus group discussion is a semi-structured group session, which in the present study aims to create an interaction between the participants and develop feasible programs to improve and control violence in such centers. The selection of this group was based on the research objective and the selected individuals were aware of hospital issues, nurses' work conditions, and hospital regulations, and could have a significant role in the implementation of violence-reduction

strategies. Three 60-90-minute focus group discussion sessions (n = 4–6) were held in the hospital's meeting room to collect data and develop strategies.

Topics presented in the focus group discussion sessions were recorded with the consent of the participants using a voice recorder. Then, the recorded materials were transcribed word by word. At the end of each session, the content analysis was used for data analysis. At the beginning of the session, a summary of findings from Phase 1, the importance of violence, and the reason behind holding the sessions were presented. At the end of each group discussion, the most important contents, noted, were read to the attendees to increase the consistency of the study (respondent-validity), and they were asked to confirm the items and correct them if necessary. In these meetings, programs and solutions to correction and improvement hospital violence were discussed, leading to decisions on feasible strategies in the center.

Purposive sampling was used to select the participants. These participants were directly or indirectly related to nursing activities and had adequate knowledge about rules and regulations. These sessions were held after obtaining permission from the hospital head. The participants were ensured about the right of withdrawal at any stage. The implementation phase was conducted over a 6-month period. A discussion of strategies is presented in the results section.

### Phase 3: Follow-up audit and post-implementation of change strategy

After implementation of the strategies, the team carried out the post-implementation audit over three weeks by utilizing the same methodology as the pre-implementation audit. In sum, 62 nurses were audited and the data were coded and analyzed into the PACES program. The reduced number of participants was due to the COVID-19 pandemic and the subsequent heavy nursing workload in different units.

## Results

### Phase 1: Baseline audit

The audit results (Fig 2) illustrated 49% compliance was reported for aggression management training (First criteria) and 8% compliance was reported for the timely support and assistance following the incident (Second criteria). Also in the third criteria, the policy in organization for risk management and safe environment had 22% compliance.

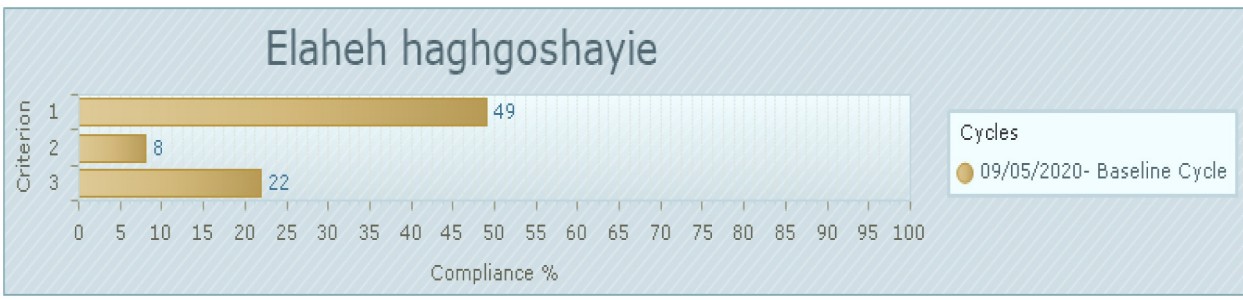

**Fig 2. Baseline compliance with the best practice for audit criteria (%).** 1. Healthcare staff receives aggression management training (42 of 86 samples taken). 2. Healthcare staff who experienced violence or aggression receives timely support and assistance following the incident (7 of 86 samples taken). 3. The health organization has a policy for risk management and ensuring a safe environment for all staff and consumers (19 of 86 samples taken).

## Phase 2: Implementation—Getting Research into Practice (GRIP)

After holding the group discussion sessions and finalizing effective strategies based on hospital conditions, rules and regulations, and organizational culture by the research team, plans and strategies were issued to the hospital units by the hospital manager. Table 3 displays the barriers, strategies, resources and outcomes.

Given risk management policies, existing written policies were not implemented properly. Therefore, the research team tried to revise these policies and add other related issues. Concerning the security guards, for example, the existing job description was not followed properly. Therefore, some sessions were held for them on their job description and the importance of following it to improve violence. Moreover, security cameras were installed in all parts of the hospital and more security guards were deployed after negotiation with the manager. The hospital management also held a meeting with the security staff to point out the importance of their profession and informed them about the hospital's expectations and improvement of hospital security. Visiting time has been an issue of the medical staff, patients, and visitors. Therefore, with the prevalence of COVID-19 in order to restrict visiting hours and prevent possible tensions, people were informed by installing a banner and on social media.

According to the same policy, security guards, did not allow visitors to meet patients in non-visiting hours. Therefore, the hospital visiting hours were restricted to one hour a day. Given reporting systems and increasing nurses' awareness of violence and related administrative processes, it was tried to develop guidelines related to reporting violence, issuing violence rules, and designing relevant posters. Staff training can prepare the skills and knowledge required to better deal with potentially violent situations. Therefore, by holding sessions on anger, communication management, training in de-escalation, it was tried to enhance their self-confidence on how to deal with violence and motivate them to give reports. Moreover, the consulting center of the faculty was coordinated to accept victims referred to them for counseling services. Also, after the event or incident occurs, the supervisors tried to hold formal and informal debriefing and competent counselling for nursing staff. Upon the decision of the management team in the hospital, a quality improvement expert was selected as the person in charge of following up the causes of violence. This person will be in direct contact with

**Table 3. Getting research into practice matrix.**

| No. | Barriers | Strategies | Resources | Outcomes |
|---|---|---|---|---|
| 1 | Unsafe work environment | • Strengthening security measures<br>• Restricted visiting hours<br>• Review of security teams' responsibilities | • Equipment like camera<br>• Banner | Increasing job satisfaction |
| 2 | Lack of reporting mechanisms and tracking violence | • Designing clear processes for workplace violence reporting.<br>• Designing instruction regarding encountering violence in the workplace system.<br>• Employee assistance programs (EAP) like providing psychological support and counseling services for victims of violent incidents | • Guidelines<br>• Poster<br>• Victims' meeting with clinical psychology | • Increasing nurses' interest to reporting workplace violence<br>• Improving nurses' confidence to resolve problems<br>• Create blame—free environment. |
| 3 | Lack of violence prevention training programs | • Training programs of communication skills, stress management and violence management to prevent violent incidents | • Invite experts about topics like assistance professors of clinical psychology for training.<br>• Translation of materials into local language<br>• JBI evidence summaries Current evidence from the literature<br>• PowerPoint presentation<br>• Brochure | • Increasing ability to identify and predicting early signs of violent behavior and management of violent incidents<br>• Create blame free environment |

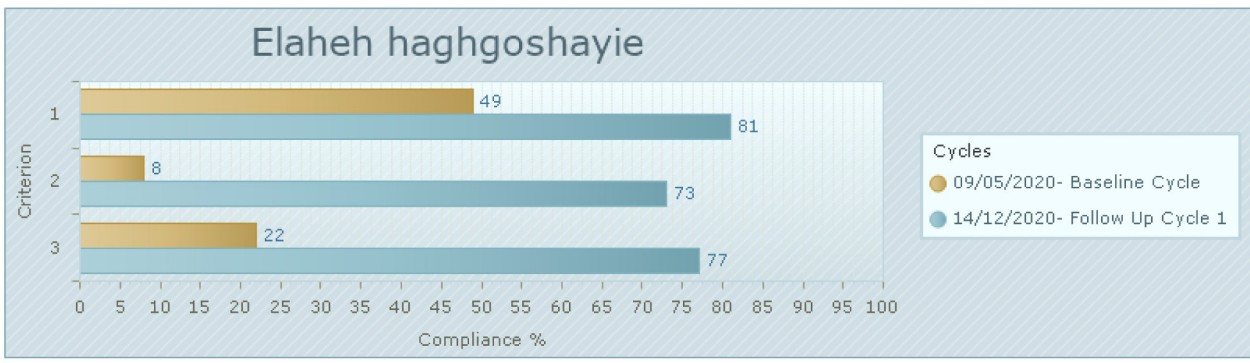

**Fig 3. Compliance with the best practice criteria in follow-up audit compared to baseline audit (%).** 1. Healthcare staff receives aggression management training. (50 of 62 samples taken). 2. Healthcare staff who experienced violence or aggression receive timely support and assistance following the incident. (45 of 62 samples taken). 3. The health organization has a policy for risk management and ensuring a safe environment for all staff and consumers. (48 of 62 samples taken).

employees who are exposed to violence. This person is responsible for the full reporting circle and is responsible for following up the results and reporting to the staff.

### Phase 3: Post-implementation audit

The results of post-implementation audit over three weeks are reflected in Fig 3. The first criterion (aggression management training) reached 81% compliance. The second criterion (timely support and assistance following the incident) improved by more than 70% and the third criterion (policy in organization for risk management and safe environment) improved from 22% to 77%.

### Discussion

This project is the first attempt to examine the current practice and implement evidence-based violence management in Shahid-Beheshti hospital in Maragheh. Baseline and follow-up data were collected to identify the barriers, strategies, and resources by using JBI PACES and GRiP tools.

The first criterion, which was based on holding violence management training courses, showed 81% compliance. It presented holding continuous training programs to use different and effective methods to deal with violence. Indeed, its factors have a significant role in violence management. Numerous studies have indicated the need for continuing education regarding violence [50–53]. For example, a study by Şenuzun and colleagues showed that 90.9% of nurses has agreed with continuing education to deal with violence and has considered it necessary [54]. Hilla and colleagues in their study showed that staff training on de-escalation is required and managers can also empower nurses through education and amending communication and adaptive skills [40].

It is found that employee flexibility and resilience can be increased through appropriate training in the workplace [51]. Also, Rahmani and colleagues found that educating nurses in terms of how to deal with patients, their communication skills and also improving the management in the healthcare system are important factors in reducing violence [45]. (However, there was no intervention in their study and it is noted only due to finding and investigating the level of violence and its related factors in a training hospital, without considering the training course for staff and evaluating its impact, as we performed). Also, the results of Stephane's

study entitled evaluation of a training program to prevent and manage patient violence in a pre-test and post-test intervention stated perceived level of various forms of violence and the level of anxiety of the nurses were significantly reduced, and self-confidence against the aggressor was increased. The results of this study showed that omega training had a positive effect, and its effect remained stable for more than 14 months after training. In addition, reducing violence affects employees' perceptions of their work; thereby, increase the quality of communication between the service provider and the patient [53]. The commitment and motivation reduce the risk of absence from work. In this regard, the evidence indicated that self-confidence, attitudes, skills, and knowledge of nurses had increased after training. However, education had no effect on reducing the incidence of violence and the reduction of the incidence of violence should be considered at the organizational level [55].

Regarding the second criterion, preparing educational pamphlets on employees' rights and preparing posters and installation was performed in the wards that showed 73% compliance. A study in Australia also found that more than 70% of healthcare workers were satisfied with the workplace violence control protocols and reporting mechanisms due to the clear policies, appropriate rules, and comprehensive reporting mechanisms [56]. Therefore, having clear instructions and awareness of related processes can lead to violence management in the hospital [50].

Another strategy used in this study was conducted to create a safe environment in the hospital (criterion 3). In line with this goal, some security cameras were purchased and installed in the hospital yard and all the sections. In this regard, Khaleghparast's study showed that that visitor's presence in wards (open visiting), more traffic and their successive questions waste personnel time and interfere with nursing care [57].

The evidence showed that the strategies to modify violence in the workplace improve security of hospital environment [2, 58]. In our project, undertaken strategies (criterion 3) have played an effective role in violence management. As it is reported the implementation of these strategies led to 77% compliance. In other words, it shows about 55% improvement.

Like other studies, there were limitations in this study. The most important limitations faced by the project team were the outbreak of COVID-19, increased workload of nurses, and their high fatigue and anxiety, which led to decrease in training courses. In addition, due to the critical situation, the sessions were held online; hence, were less effective than face-to-face training. Additionally, we faced internet problems and low speed in holding the sessions. In this study, the target group was only nurses. It is better than other clinical groups and patients to be considered in the future research. Finally, due to the selection of a hospital, the results of the study will not necessarily be generalizable to other hospitals and departments. Despite these limitations, this study presented significant improvements in evidence-based violence management among nurses.

The findings of this study create important insights into the management of evidence-based violence in Shahid-Beheshti Hospital. This study is a promising start for health policy makers because violence and the threat of violence are undoubtedly major stressors for nurses and barriers to effective patient care. Additionally, the results of this study provide the positive direction into implementing evidence-based violence management in other hospitals of Maragheh University of Medical Sciences. This project has the potential to raise awareness amongst nurses and hospital managers on the facilitators of the violence management among nurses and improvement strategies in Iranian hospitals.

## Conclusion

Based on the objectives of this study, best available evidence and design strategies include holding training classes on violence management, communication skills, creating a safe

environment in the hospital and communicating guidelines for dealing with violence at work. They increase nurses' awareness, change performance, and increase their resilience. Ultimately, evidence-based violence management was undertaken. In the future, this study should audit other clinical and administrative staff, patients and their companions by taking a great step towards reducing violence in the hospital by implementing evidence-based strategies. This study can be a basis for starting evidence-based studies in the field of violence management in the Iranian hospitals. In addition, by providing better evidence about the prevalence of workplace violence, it is possible to motivate the managers of healthcare organizations to intervene in workplace violence by improving the management of violence in health care systems in three dimensions of prevention, intervention in workplace violence, and follow-up of workplace violence cases to take more effective steps.

## Acknowledgments

Research support was provided by Research Center for Evidence-Based Health Management at Maragheh University of Medical Sciences. In addition, the authors would like to thank Research Centre for Evidence Based Medicine at Tabriz University of Medical Sciences for supporting.

## Author Contributions

**Conceptualization:** Ghasem Abedi, Edris Hasanpoor.

**Data curation:** Elaheh Haghgoshayie, Edris Hasanpoor.

**Formal analysis:** Ghasem Abedi, Elaheh Haghgoshayie, Edris Hasanpoor.

**Investigation:** Ghasem Abedi, Morteza Nazari.

**Methodology:** Elaheh Haghgoshayie, Razieh Vejdani.

**Project administration:** Elaheh Haghgoshayie.

**Resources:** Elaheh Haghgoshayie.

**Software:** Elaheh Haghgoshayie.

**Supervision:** Elaheh Haghgoshayie, Edris Hasanpoor.

**Validation:** Ghasem Abedi.

**Visualization:** Edris Hasanpoor, Morteza Nazari, Razieh Vejdani.

**Writing – original draft:** Ghasem Abedi, Elaheh Haghgoshayie, Jalil Etemadi, Morteza Nazari, Razieh Vejdani.

**Writing – review & editing:** Ghasem Abedi, Edris Hasanpoor, Jalil Etemadi.

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
