## [Decision Letter · Decision Letter 0]

19 Sep 2022

PONE-D-22-15707Improvement of violence management among nurses in Iran: a best practice implementation projectPLOS ONE

Dear Dr. Hasanpoor,

Thank you for submitting your manuscript to PLOS ONE. After careful consideration, we feel that it has merit but does not fully meet PLOS ONE’s publication criteria as it currently stands. Therefore, we invite you to submit a revised version of the manuscript that addresses the points raised during the review process.

 Please see the comments from one reviewer below. Please note that we have only been able to secure a single reviewer to assess your manuscript. We are issuing a decision on your manuscript at this point to prevent further delays in the evaluation of your manuscript. Please be aware that the editor who handles your revised manuscript might find it necessary to invite additional reviewers to assess this work once the revised manuscript is submitted. However, we will aim to proceed on the basis of this single review if possible.  In addition to the reviewer comments, we also request the following:

1) Please ensure that the manuscript is thoroughly copyedited before resubmission

2) Please clarify how participants provided consent. We note that you have stated that they completed a verbal consent form - please clarify whether the form was written or verbal, and if verbal, how consent was witnessed.

3) Please state in your cover letter whether all the members of the team listed in Table 1 have agreed to be named in the manuscript.

We look forward to receiving your revised manuscript.

Kind regards,

Hanna Landenmark

Staff Editor

PLOS ONE

Journal Requirements:

2. Thank you for sending us the data set underlying the results presented in your PLOS ONE submission. We notice that some of the information included in the data set may be potentially identifying. Please ensure that the data shared are in accordance with participant consent and provide only the data that are used in this specific study. To ensure patient confidentiality, we would recommend removing the columns with the names in, unless these team members have all agreed to be named. Additional guidance on preparing raw clinical data for publication can be found in our Data Policy FAQs (https://journals.plos.org/plosone/s/data-availability#loc-clinical-data).

4. Please ensure that you refer to Figure 1 in your text as, if accepted, production will need this reference to link the reader to the figure.

Reviewers' comments:

Reviewer's Responses to Questions

**Comments to the Author**

1. Is the manuscript technically sound, and do the data support the conclusions?

Reviewer #1: Yes

2. Has the statistical analysis been performed appropriately and rigorously? 

Reviewer #1: N/A

3. Have the authors made all data underlying the findings in their manuscript fully available?

Reviewer #1: No

4. Is the manuscript presented in an intelligible fashion and written in standard English?

Reviewer #1: No

5. Review Comments to the Author

Reviewer #1: This study aims to identify and provide strategies against workplace violence among nurses in a single small-scale hospital in Iran. The topic is important in practice. It's strength is including pre- and post- intervention evaluations, and the policy implementation was performed in real-life workplace. I have a few minor comments and I hope the authors will find them useful for improving this manuscript.

1. I suggest the authors to trim the introduction section because now it looks like a project report rather than a scientific report. Please consider shorten the sections introducing definitions of workplace violence and its prevalence and focus on the knowledge gap of implementating strategies and policies for workplace violence prevention.

2. One of the limitations is generalizability of the results. I wonder how such implementation project could be applied to a larger scale healthcare facility where there is thousands of employees? Please elaborate.

3. The pre-intervention audit was performed between May and July 2020. When did the post-intervention audit take place? Would the time interval matter?

4. In page 9, "the project steps are shown in Figure 2". I suppose this should be Figure 1. Table 3 was not cited in the text.

5. There are some typos and grammar errors. The manuscript will benefit from English-editing.

6. PLOS authors have the option to publish the peer review history of their article (what does this mean?). If published, this will include your full peer review and any attached files.

Reviewer #1: No

---

## [Author Response · Author response to Decision Letter 0]

3 Jan 2023

Reviewer #1: This study aims to identify and provide strategies against workplace violence among nurses in a single small-scale hospital in Iran. The topic is important in practice. Its strength is including pre- and post- intervention evaluations, and the policy implementation was performed in real-life workplace. I have a few minor comments and I hope the authors will find them useful for improving this manuscript. 

1. We notice that your manuscript file was uploaded on May 31 2022. Please can you upload the latest version of your revised manuscript as the main article file, ensuring that does not contain any tracked changes or highlighting. 

Both of highlighting manuscript and main manuscript were attached .

2. Please include a separate legend for Figure 1 in your manuscript.

It was revised. 

3. Please ensure that you refer to Table 3 in your text as, if accepted, production will need this reference to link the reader to the Table

It was referred in text.

---

## [Editor Report · Decision Letter 1]

10 Apr 2023

Improvement of violence management among nurses in Iran: the best practice implementation project in a health promoting hospital

PONE-D-22-15707R1

Dear Dr. Haghgoshayie,

We’re pleased to inform you that your manuscript has been judged scientifically suitable for publication and will be formally accepted for publication once it meets all outstanding technical requirements.

Kind regards,

José J. López-Goñi

Academic Editor

PLOS ONE
---

## [Editor Report · Acceptance letter]

13 Nov 2023

PONE-D-22-15707R1 

Improvement of violence management among nurses in Iran: the best practice implementation project in a health promoting hospital 

Dear Dr. Haghgoshayie:

I'm pleased to inform you that your manuscript has been deemed suitable for publication in PLOS ONE. Congratulations! Your manuscript is now with our production department. 

Kind regards, 

on behalf of

Dr. José J. López-Goñi 

Academic Editor

PLOS ONE